# Recovery of Biophenols from Olive Vegetation Waters by Carbon Nanotubes

**DOI:** 10.3390/ma15082893

**Published:** 2022-04-14

**Authors:** Pierantonio De Luca, Anastasia Macario, Carlo Siciliano, Janos B.Nagy

**Affiliations:** 1Dipartimento di Ingegneria Meccanica, Energetica e Gestionale, University of Calabria, I-87036 Arcavacata di Rende, Italy; janos.bnagy1@gmail.com; 2Dipartimento di Ingegneria per l’Ambiente, University of Calabria, I-87036 Arcavacata di Rende, Italy; anastasia.macario@unical.it; 3Dipartimento di Farmacia e Scienze della Salute e della Nutrizione, University of Calabria, I-87036 Arcavacata di Rende, Italy; carlo.siciliano@unical.it

**Keywords:** adsorbtion, biophenols, carbon nanotubes, olive vegetation water, polyphenols

## Abstract

In this work, the possibility of using carbon nanotubes for the treatment of olive vegetation waters (OVWs) was investigated. In general, the disposal of OVWs represents an important environmental problem. The possibility of considering these waters no longer just as a problem but as a source of noble substances, thanks to the recovery of biophenols from them, was tested. In particular, predetermined quantities of olive vegetation waters were treated with carbon nanotubes. The quantities of adsorbed biophenols were studied as a function of the quantities of carbon nanotubes used and the contact time. The experimental conditions for obtaining both the highest possible quantities of biophenol and a purer adsorbate with the highest percentage of biophenols were studied. After the adsorption tests, the vegetation waters were analyzed by UV spectrophotometry to determine, in particular, the variation in the concentration of biophenols. The carbon nanotubes were weighed before and after each adsorption test. In addition, kinetic studies of the adsorption processes were considered. Carbon nanotubes proved their effectiveness in recovering biophenols.

## 1. Introduction

The production of olive oil results in the production of by-products that are particularly dangerous for the environment. The main by-products are pomace and so-called olive vegetation waters (OVWs).

The pomace appears as a substantially solid phase and is made up of various organic compounds (such as cellulose, carbohydrates, proteins, phenolic and inorganic compounds, etc.) and parts of the skin and olive stones [1]. Pomace is generally recycled for the production of animal feed, for the production of biogas or for the production of compost after appropriate treatment [2]. The olive vegetation waters, on the other hand, represent the liquid by-products of oil production. The nature of the pomace and the olive vegetation waters is very variable and depends mainly on the processing process and on the geographical and climatic characteristics of the place where the olive tree is grown. As with pomace, disposal is also an important problem for olive vegetation waters since the latter consist mainly of an important organic fraction, fundamentally phenolic in nature, which is poorly biodegradable [3]. They are also characterized by a high acidity. Massive release of OVWs into the environment can involve the modification or destruction of the bacterial flora of the soils on which they are released or the pollution of aquifers, with serious consequences for the environment or for the purification systems of civil waters [4,5]. Considering that millions of tons of such waters are produced every year in the short olive pressing season, the problem of disposal is of great interest. These by-products, although their disposal is a problem, if properly treated, can represent a source of particularly valuable substances. Vegetable waters, for example, are rich in biophenols [3,6].

It is known that biophenols are excellent allies for human health [7,8]. In fact, they have antioxidant properties [9,10,11]. Many studies have confirmed the ability of these substances to act beneficially on cardiovascular diseases and inflammations [12,13,14]. Recent research also confirmed the antitumor action of these molecules [15]. Thanks to these properties, biophenols are generating more and more interest in the pharmaceutical industry; therefore, research is increasingly aimed at optimizing and researching new recovery methods. As is known, the traditional methods of recovery and treatment of contaminated water are those of centrifugation, membrane, photocatalysis, reverse osmosis, ultrafiltration and adsorption [16,17,18].

Carbon nanotubes (CNTs) have proved to be particularly important materials, endowed with marked versatility and applicability in various sectors, i.e., as fiber reinforcers [19], adsorbents [20,21,22], electronic sensors [23,24,25], etc. These materials have been shown to have a high competitive adsorbent capacity with the more classic adsorbent materials such as chitosans [26,27], membranes [28,29,30], zeolites [31,32,33] and zeotopes [34,35,36].

They consist of rolled-up graphene sheets that form tubes, the diameter of which is nanometric in size. Thanks to this high-adsorbing capacity, carbon nanotubes are advantageously used for the purification of water from organic substances [22,37,38], dyes [39,40], gasoline [41], etc.

To our knowledge, there are no studies reported in the literature that used carbon nanotubes for the treatment of olive vegetation waters. Therefore, in this experimental work, carbon nanotubes for the recovery of biophenols from the olive vegetation waters of the olives were studied and tested.

In addition to biophenols, OVWs are rich in various organic substances, such as organic acids, reducing sugars, sugar alcohols, etc., and minerals such as potassium, phosphorus and calcium. For this reason, the recovery of biophenols from OVWs is not always a simple and easy operation. Taking this aspect into consideration, this research aims to define the best operating conditions to give a high selectivity in the adsorption of biophenols from OVWs using carbon nanotubes.

## 2. Materials and Methods

### 2.1. Materials

#### 2.1.1. Carbon Nanotubes

Unpurified carbon nanotubes (CNTs) were used. These were synthesized by catalytic chemical vapor deposition technique (CCVD) and extensively characterized as reported in previous papers [38,39,40]. In short, the nanotubes used had a purity of 95%. They had an average pore width of 103.70 Angstroms and a specific BET area of 108.70 m^2^/g.

#### 2.1.2. Olive Vegetation Waters (OVWs) and Their Pre-Treatment

The OVWs were supplied by an oil mill of a local, private company. Before being subjected to adsorption tests, these samples were previously treated. Initially, the olive vegetation waters were subjected to centrifugation for a time of 10 min and at a rotation speed of 13,000 rpm.

Subsequently, the OVWs were separated from the precipitate and subjected to vacuum filtration with a borosilicate glass filter funnel (Pyrex, Corning, NY, USA, porosity 5), with the aim of eliminating even the finest, suspended particles.

The liquid phase thus obtained, the OVWs were stored in the refrigerator at a temperature of 4 °C before being analyzed and subjected to adsorption tests.

### 2.2. Adsorption Tests

The adsorption tests were conducted in discontinuous mode and on different systems consisting of a constant volume of OVWs equal to 20 mL and different quantities of carbon nanotubes equal to 0.3, 0.5 and 1.0 g, respectively, called systems 1, 2 and 3. The adsorption times were 5, 10, 20, 45, 60 and 120 min, respectively. The volume of OVWs used for each single test was 20 mL; this remained unchanged for all the tests. Conversely, the quantity of CNTs entered varied. The OVWs/CNTs system was stirred at a constant speed of 250 rpm for characteristic times in a well-sealed container to avoid evaporation phenomena.

#### After Adsorption Tests

Subsequently, at the end of the programmed stirring time, the CNTs were separated by filtration by means of a vacuum pump. The filter used was a borosilicate glass filter funnel (Pyrex, porosity 5).

The filtered phase was collected and stored at 4 °C and analyzed by UV spectrophotometer while the solid part, which remained on the paper filter, was placed to dry in the oven at 100 °C for about 12 h and weighed before and after to evaluate the increase in weight.

### 2.3. Characterization

The UV-3100 Shimadzu spectrophotometer (Kyoto, Japan) was used to characterize the OVWs after the adsorption tests. The analyses were referred to the characteristic band at 280 nm. Generally, measurement of biophenols by UV spectrophotometry is referred at 280 nm, which represents a suitable compromise as most phenols absorb considerably at this wave-length [42,43]. The measurements were repeated three times, and their average was taken into account.

## 3. Results and Discussion

### 3.1. OVWs Characterization

The following Figure 1 shows the characteristic UV band relative to the initial OVWs sample before being subjected to adsorption tests.

It is known that, in OVWs, there are about twenty different types of biophenol, among which, the most abundant is hydroxytyrosol, and their composition is different from that present in olives. In the OVWs considered in this research, a well-defined band around 280 nm was observed, which represents a reference for the presence of biophenols in OVWs in agreement with other works reported in the literature [42,43]. The initial concentration of biophenols in the OVWs was found to be 7.4 g/L.

### 3.2. Total Mass Adsorbed after the Tests

Figure 2 shows the total mass adsorbed by the carbon nanotubes after the adsorption tests carried out on the vegetation waters as a function of the contact time and the quantity of nanotubes used.

The data shown in Figure 2a–c show, in all cases, a considerable amount of mass adsorbed by the carbon nanotubes, which highlights an important, purifying capacity of the latter towards OVWs. In general, the total mass is attributable to the adsorption of the numerous chemical species present in the OVWs [44]. It is clear that the amount of adsorbed material is strictly dependent on the contact time and the number of nanotubes used.

As can be seen in Figure 2a, in the system that used only 0.3 g of carbon nanotubes, after only 5 min, there was an amount of adsorbed mass of about 0.4 g, which corresponded to an increase in the weight of the nanotubes of about 35%. After 60 min, the adsorbed quantity was 0.55 g, which corresponded to an increase in the weight of the nanotubes of about 80%.

Figure 2b shows the amount of total mass adsorbed in the system using 0.5 g of carbon nanotubes. The adsorbed quantity at 5 min was about 0.66 g and 1.2 g after 45 min, which corresponded to an increase in the weight of the carbon nanotubes of about 140%.

Figure 2c shows the total amount of mass adsorbed in the system using 1 g of carbon nanotubes. In this case, the dependence of time was less evident. In fact, already in the first 5 min, the adsorbed mass was about 4.8 g, which corresponded to an increase in the weight of the nanotubes of about 375% and then remained more or less constant, reaching an adsorbed quantity of about 5.08 g at 60 min, which corresponded to a weight increase of 400%.

The high purifying action of carbon nanotubes against OVWs was also visible visually. Figure 3, as an example, shows the images of the olive vegetation water samples relating to the system that used 1 g of carbon nanotubes as a function of the adsorption time. After only 45 min, an important clarification of the OVWs could be observed with the naked eye.

### 3.3. Amount of Biophenols Retained after Adsorption Tests

In the continuation of the research, only the amount of biophenols adsorbed during the tests was measured. For this purpose, the biophenols concentrations were measured before and after the tests in the OVWs by UV spectrophotometry. As reported in Section 3.1, the initial concentration of biophenols in the OVWs was equal to 7.4 g/L. Specifically, 0.148 g of biophenols were initially present in the 20 mL of OVWs used for each system. The following Figure 4a–c reports the grams of biophenols adsorbed as a function of time for systems with three different quantities of nanotubes.

Figure 4a–c shows an important adsorbing action of the carbon nanotubes against the biophenols present in the OVWs. The adsorbed quantities are a function of the contact time and the quantities of nanotubes used. In all three cases, as the contact time increased, the quantities of biophenols adsorbed increased until they stabilized at about 60 min of adsorption. The same linearity was not found for the quantities of biophenols adsorbed as the quantities of nanotubes used increased. In fact, an increase in nanotubes from 0.3 to 0.5 g was accompanied by an increase in the quantities of biophenols adsorbed of 0.065 and 0.088 g, respectively, after 60 min. On the other hand, when the quantity of nanotubes used was further increased to 1 g, after a contact time of 60 min, the grams of adsorbed biophenols were lower compared to the system with 0.5 carbon nanotubes, which became equal to 0.076.

The data reported in the previous Section 3.1 showed that, by increasing the quantity of nanotubes used, there was an increase in the total quantity of material adsorbed by the OVWs. The same was not true if the adsorption of biophenols alone was considered. It is evident that, given the large number of chemical species present in the OVWs, competition phenomena arose. An increase in the quantities of nanotubes beyond a certain value generated agglomeration phenomena of the nanotubes due to poor dispersion of the latter, visible to the naked eye a few seconds from the start of the adsorption tests. This situation predisposed the selection towards the adsorption of more mobile species that are able to reach the adsorption sites more easily. It is, therefore, important to evaluate the most appropriate quantities to use. In the cases studied, the use of 0.5 g of carbon nanotubes on 20 mL of OVWs was the most advantageous as it offered the greatest amount of biophenols adsorbed at 60 min.

### 3.4. Percentage of Biophenols in the Total Adsorbed Mass

The following Figure 5a–c reports the percentages of biophenols with respect to the total mass retained for all three systems studied.

The data obtained showed that, by increasing the quantity of nanotubes in the systems, the percentage of biophenols with respect to the total adsorbed mass decreased significantly. In fact, taking into consideration, for example, a time of 120 h, the percentages were 12.7, 7 and 1.5, respectively, for systems with 0.3, 0.5 and 1.0 g of carbon nanotubes. This, once again, confirms the hypothesis that greater quantities of carbon nanotubes do not favor the adsorption of biophenols due to agglomeration phenomena that favor the adsorption of more mobile and less sterically hindered species. Another aspect that could be observed was that, as the quantity of nanotubes increased, the influence of time was less evident until it was almost zero, as in the case of the system that used 1.0 g of nanotubes. System 3, which used the highest quantities of nanotubes, equal to 1 g, reported the lowest percentage of biophenols with minimal variability with respect to the adsorption time. The following Table 1 reports the % removal by weight with respect to the amount of biophenols initially present in the 20 mL of OVWs, equal to 0.148 g, in the various systems studied as a function of the adsorption time.

At this point, an important consideration and distinction must be made with respect to the absolute quantity of biophenols adsorbed and the percentage of biophenols with respect to the total mass adsorbed. It is clear that the recovery of biophenols from OVWs presupposes the need to adsorb the highest possible amount. However, it is also necessary to obtain an adsorbate as pure as possible; that is, an adsorbate with a high percentage of biophenols. It is, therefore, appropriate to find a compromise between both needs. The data presented so far show that the highest amount of biophenols absorbed was obtained with system 2, containing 0.5 g of carbon nanotubes, and with an adsorption time of 60 min. In this case, the grams of adsorbed biophenols were equal to about 0.09 g, which corresponded to a percentage removal compared to the initial grams of biophenols present in the OVWs equal to 59.8% (Table 1). However, the percentage of biophenols with respect to the total adsorbed mass was approximately 7.25% (Figure 5b).

The purest product, i.e., with a percentage of biophenols higher than the total adsorbed mass, was obtained for system 1, with 0.3 g of carbon nanotubes and for a reaction time of 60 min equal to 12% (Figure 5a) against a percentage removal equal to 43.48% (Table 2).

Therefore, the conditions that give an adsorbate with the highest percentage of biophenols are the use of system 2 with a time of 60 min, while a purer adsorbate is obtained with system 1 with a time of 60 min.

From the data obtained, it was possible to hypothesize a treatment of 1 L of OVWs in order to highlight more concretely which are, in the best experimental conditions found, the quantities of carbon nanotubes that must be used and the quantities of biophenols that are recovered.

The following Table 2 reports the grams of carbon nanotubes to be used, grams of adsorbed biophenols, grams of total adsorbate and the ratio, g_biophenol ads._/g_total ads._, assuming a treatment of 1 L of OVWs and operating with the conditions relating to systems 1, 2 and 3.

From data reported in Table 2, it is clear that using only 15 g of carbon nanotubes and in 60 min, it is possible to recover 0.88 g of biophenols from 1 L of OVWs. Data obtained are certainly interesting considering also that the carbon nanotubes, after a regeneration, for example, with desorption in organic solvents, can be reused again. This makes the method simple and, above all, economically advantageous.

### 3.5. Study on Reaction Kinetics

The following Table 3 reports the adsorption rates, expressed as grams of adsorbed biophenols/minute, for the three systems studied in different time intervals: 0–5, 5–60 and 60–120 min.

The data reported show that, in the first time interval, between 0–5 min, the adsorption rate increased linearly as the amount of nanotubes increased. This agreed with a large availability of nanotubes which, in the first interval, were still far from saturation and also from agglomeration phenomena.

In the interval between 5–60 min between the increase of nanotubes from 0.3 to 0.5, it did not lead to an increase in the adsorption rate. Furthermore, a decrease in speed was evident for the higher quantity of nanotubes (1 g). Both cases confirmed the previously exposed hypothesis of agglomeration formation and selectivity towards more mobile species than biophenols.

After 60 min, in the interval between 60–120, the nanotubes reached a state of saturation, and the speed returned to very low values for all the systems studied.

## 4. Conclusions

The results obtained allow us, first of all, to state that carbon nanotubes can be advantageously used for the recovery of biophenols from the vegetation waters of the olives.

In addition to biophenols, carbon nanotubes simultaneously adsorb other substances, as the vegetation waters of olives are rich in various components.

In the recovery of biophenols from OVWs through carbon nanotubes, it is necessary to aim to obtain both an absolute highest possible quantity of adsorbed biophenols and an adsorbate that has the highest possible percentage of biophenols.

The studies carried out showed that the quantities of carbon nanotubes used play a fundamental role with respect to the previous aspects.

The reported data allow us to conclude that, to obtain a purer adsorbate that is richer in biophenols, it is necessary to opt for a low quantity of carbon nanotubes.

An increase in the quantities of nanotubes generates agglomeration phenomena of carbon nanotubes due to poor dispersion of the latter. This situation predisposes the selection towards the adsorption of more mobile species that are able to reach the adsorption sites more easily.

In summary, it is possible to state that a greater adsorption selectivity with respect to biophenols in OVWs is guaranteed by an adequate and weighted quantity of carbon nanotubes. An excess of the quantities of carbon nanotubes presupposes a lowering of selectivity.

## Figures and Tables

**Figure 1 materials-15-02893-f001:**
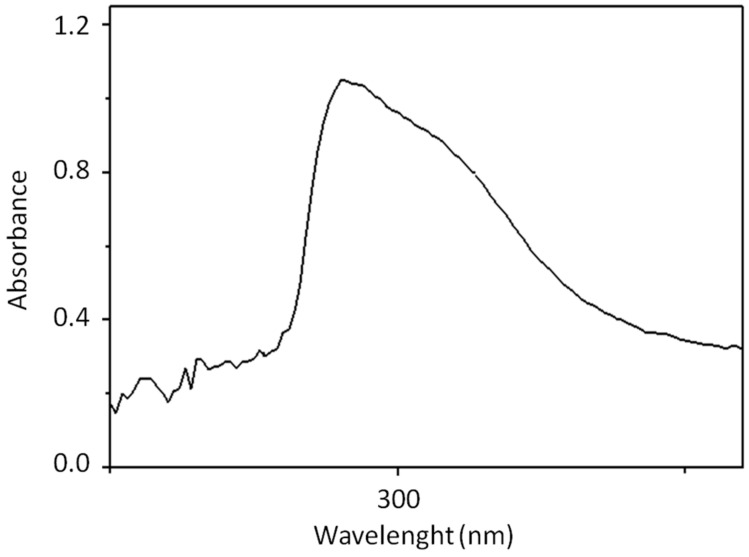
Characteristic UV band of an OVWs sample.

**Figure 2 materials-15-02893-f002:**
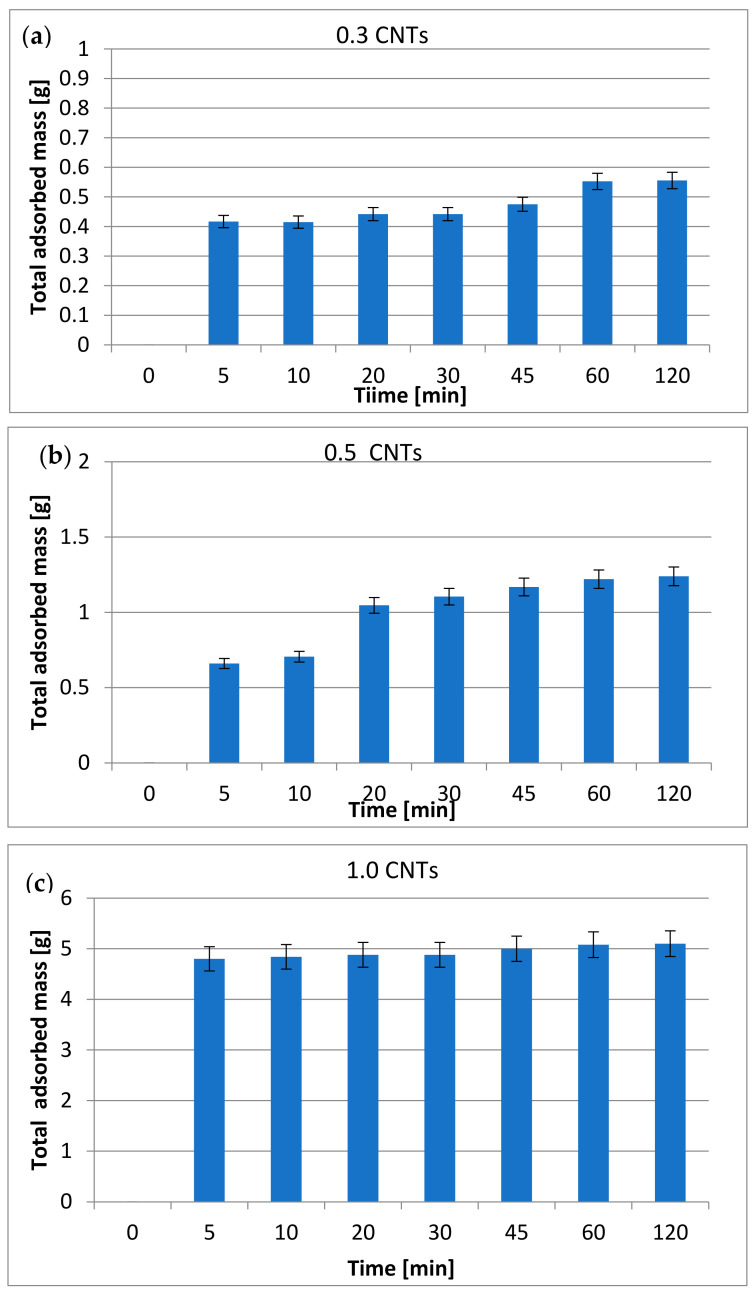
Total adsorbed mass of the carbon nanotubes after the adsorption tests as a function of the treatment times for systems 1, 2 and 3, respectively, characterized by (**a**) 0.3, (**b**) 0.5 and (**c**) 1.0 g of carbon nanotubes.

**Figure 3 materials-15-02893-f003:**
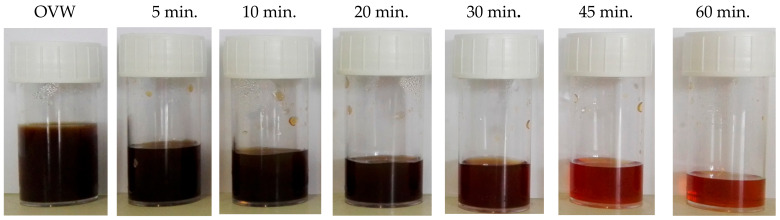
Samples of olive vegetation waters at different adsorption times in the system consisting of 20 mL of OVWs and 1 g CNT.

**Figure 4 materials-15-02893-f004:**
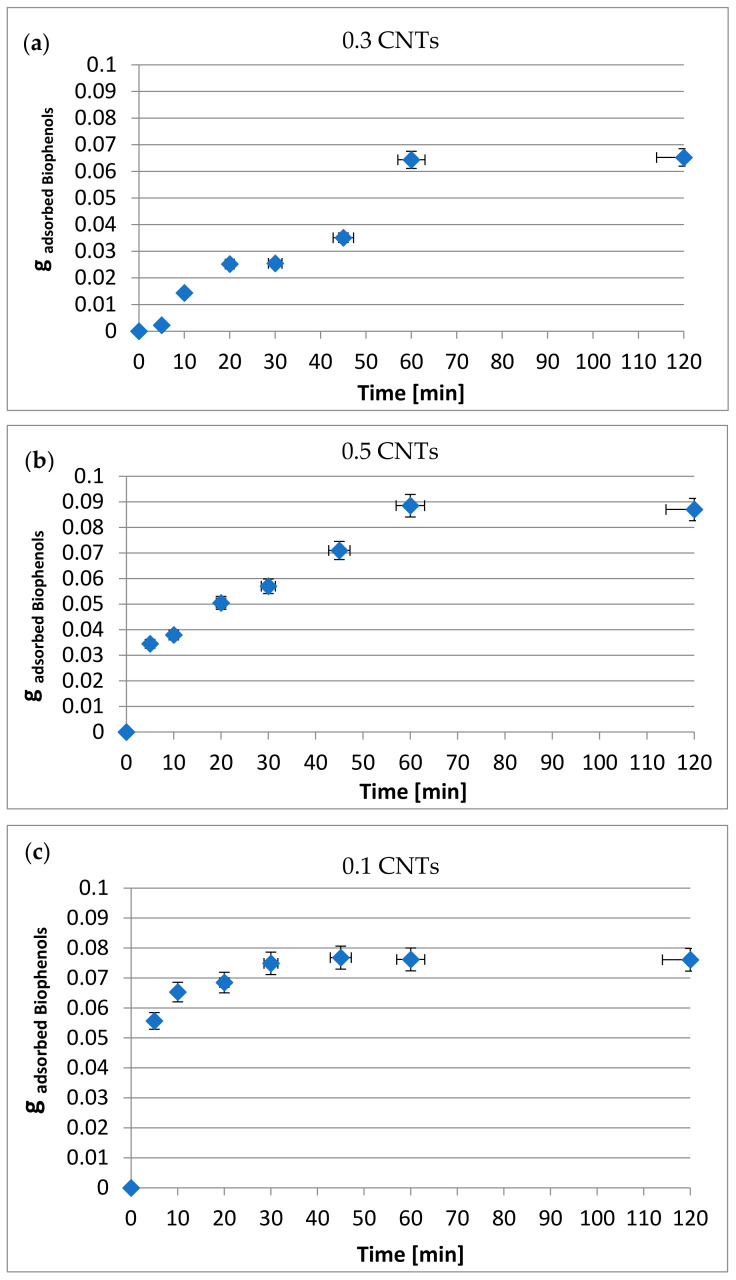
Quantity of adsorbed biophenols as a function of time in systems with (**a**) 0.3, (**b**) 0.5, (**c**) 1.0 g of nanotubes in 20 mL of OVWs.

**Figure 5 materials-15-02893-f005:**
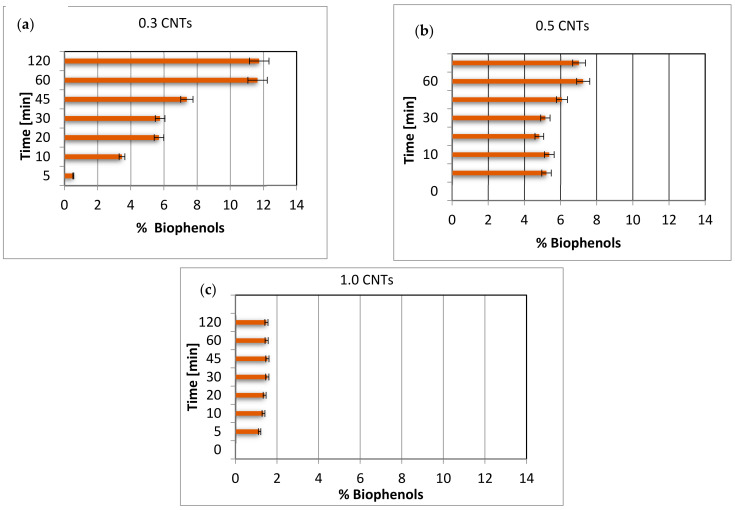
Percentage of biophenols with respect to the total adsorbed mass for systems with (**a**) 0.3, (**b**) 0.5, (**c**) 1.0 g of carbon nanotubes.

**Table 1 materials-15-02893-t001:** Percentage removal of biophenols with respect to the initial quantity of polyphenols present in the OVWs.

	Recovery [%]		
Time [min]	System 1 (0.3 CNTs)	System 2 (0.5 CNTs)	System 3 (1.0 CNTs)
5	1.52	23.31	37.63
10	9.73	25.65	44.12
20	17.03	34.12	46.28
30	17.23	38.51	50.61
45	23.72	47.97	51.90
60	43.48	59.80	51.49
120	44.09	58.78	51.42

**Table 2 materials-15-02893-t002:** Grams of carbon nanotubes to be used, g_biophenols adsorbed_, g_adsorbate total_ and the ratio g_biophenol ads_/g_total adsorbate_, assuming a treatment of 1 L of OVWs and operating with the conditions relating to systems 1, 2 and 3 at 60 min as contact time.

System	Time [min]	CNTs [g]	Biophenols Adsorbed [g]	Adsorbate Total [g]	g_biophenol ads_/g_total adsorbate_
1	60	15	0.88	2.02	0.436
2	60	25	4.42	61	0.072
3	60	50	3.81	254	0.015

**Table 3 materials-15-02893-t003:** Adsorption rate (g/min) in different time intervals for systems with different contents of carbon nanotubes.

System	Range Adsorption Time		
	0–5 [min]	5–60 [min]	60–120 [min]
(1) 0.3 g_CNTs_/20 mL _OVWs_	0.0005	0.001	2 × 10^−5^
(2) 0.5 g_CNTs_/20 mL _OVWs_	0.0069	0.001	2 × 10^−5^
(3) 1.0 g_CNTs_/20 mL _OVWs_	0.0111	0.0003	2 × 10^−6^

## Data Availability

Data are contained within the article.

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
