# Peer review of "Recovery of Biophenols from Olive Vegetation Waters by Carbon Nanotubes"

_materials, 2022, doi:10.3390/ma15082893_

Round 1

Reviewer 1 Report

  1. The possibility of using carbon nanotubes for the treatment of olive vegetation waters was investigated in this manuscript. It is interesting and there may have some development in the application of the carbon nanotubes. However, there are some question (experimental design, data analyzing, text for easy reading and attracting, and the format) presents in the current version of this manuscript, then I give some suggestion as bellowing:

    1. Normally, the number of references should not be over 3. Then, the reference 19-25 and 28-36 should be simplified.
    2. Line 62-71 should be merged in one paragraph, and all the property should be expressed out and don’t use the etc.
    3. The aim of this research is also needed, would you please add it at the end of the paragraph (Line 72-75). It is better if the current Question/problem exited is clearly explained.
    4. Line 79 and 80: the materials should be clearly explained, such as the manufacture, size, property, etc.
    5. Line 88 to 92: suggest to form in one paragraph.
    6. Please Change Table 1 into one sentence.
    7. Line 106 to 113: suggest to form in one paragraph.
    8. For section 2: how many times the test repeated and the data analyzing method is absent, could you please add them.
    9. For section 3.1: the result of OVWs is advised to be compared and discussed with other researches.
    10. For section 3.2 to 3.5: the data should be analyzed, for example ANOVA, regression, and compare result with other results.
    11. All the three figures in Fig. 5 are suggested to be set in one line or in one figure.
    12. For section 3: the aim/destination of this research (fulfilled or not), shortcoming, suggestion to the following research, and the prospect for the carbon nanotubes using in adsorption of biophenols is encourage to be added.
    13. Table 4 in section 4 is suggested to be moved to section 3.
    14. For section 4: some problems present in this section, such as containing results, discussion; then I suggest the author rewrite the conclusion in one paragraph, which is different from the abstract.

Author Response

Dear colleague,

            first of all we thank you for the time you have committed to reviewing our work.

We have taken your suggestions into consideration and made adequate changes to the manuscript and answered and clarified our reasons.

Below are our answers following your suggestions (in the manuscript the changes are reported in yellow color).

Best regards

The authors

    1. Normally, the number of references should not be over 3. Then, the reference 19-25 and 28-36 should be simplified.

R.1) The references have been distinguished and simplified.

    1. Line 62-71 should be merged in one paragraph, and all the property should be expressed out and don’t use the etc.

R.2) We are not sure if we understood your suggestion well. If we have understood your suggestion well, we prefer, if this is not important to you, to leave this part in the introduction to avoid changing the organization of the manuscript much.

    1. The aim of this research is also needed, would you please add it at the end of the paragraph (Line 72-75). It is better if the current Question/problem exited is clearly explained.

R.3)The following sentence was added in the paragraph 1:“In addition to biophenols, OVWS are rich in various organic substances such as organic acids, reducing sugars, sugar alcohols, etc., and minerals such as potassium, phosphorus, calcium. For this reason, the recovery of biophenols from OVWs is not always a simple and easy operation. Taking this aspect into consideration, this research aims to define the best operating conditions to have a high selectivity in the adsorption of biophenols from OVW using carbon nanotubes..”.

    1. Line 79 and 80: the materials should be clearly explained, such as the manufacture, size, property, etc.

R.4) The following sentence was added: “In short, the nanotubes used have a purity of 95%. They have an average pore width of 103.70 Angstroms and a specific BET area of 108.70 m2 /g.”

    1. Line 88 to 92: suggest to form in one paragraph.

R.5) We have changed the title of the paragraph in “ 2.1.2 Olive vegetation waters (OWVs) and their pre-treatment”.

    1. Please Change Table 1 into one sentence.

R.6) Table 1 was removed from the manuscript. Also in paragraph 2.2 the following sentence has been added: “The adsorption tests were conducted in discontinuous mode and on different systems consisting of a constant volume of OVWs equal to 20mL and different quantities of carbon nanotubes equal to 0.3; 0.5 and 1.0 grams, respectively called system 1, 2 and 3. The adsorption times were 5, 10, 20, 45 ,60, 120 minutes.”

    1. Line 106 to 113: suggest to form in one paragraph.

R.7) The lines 106 -113 block has been distinguished in a new paragraph called. : 2.2.1 After adsorpition tests.

    1. For section 2: how many times the test repeated and the data analyzing method is absent, could you please add them.

R.8)This sentence was added in paragraph 2.3 “The measurements were repeated three times and their average was taken into account”.

    1. For section 3.1: the result of OVWs is advised to be compared and discussed with other researches.

R.9)This sentence was added in paragraph 3.1:“ It is known that in OVWs there are about twenty different types of biophenols, among which the most abundant is hydroxytyrosol, and their composition is different from that present in olives. In the OVWs considered in this research, a well-defined band around 280nm was observed, which represents a reference for the presence of biophenols in OVWs in agreement with other works reported in the literature [44,45].”

    1. For section 3.2 to 3.5: the data should be analyzed, for example ANOVA, regression, and compare result with other results.

R.10)Error bars have been added to the graphs. We are unable to compare our results with others as we are unaware of other works that have used carbon nanotubes for the treatment of OVWs.

    1. All the three figures in Fig. 5 are suggested to be set in one line or in one figure.

R.11) We tried to put the figures in one line but this kind of a bad resolution. If this does not create a problem, we prefer to leave Figure 5 as it stands.

    1. For section 3: the aim/destination of this research (fulfilled or not), shortcoming, suggestion to the following research, and the prospect for the carbon nanotubes using in adsorption of biophenols is encourage to be added.

R.12) The following sentences have been added in section 3.4:“ Data obtained are certainly interesting, considering also that the carbon nanotubes after a regeneration, for example with desorption in organic solvents, can be reused again. This makes the method used simple and above all economically advantageous.

    1. Table 4 in section 4 is suggested to be moved to section 3.

R.13)Table 4 (now Table 2) has been moved to paragraph 3.4. The following sentences have been added in section 3: ”From the data obtained it was possible to hypothesize a treatment of 1 L of OVWs in order to highlight more concretely which are, in the best experimental conditions found, the quantities of carbon nanotubes that must be used, and the quantities of biophenols that are recovered. The following Table 2 reports the grams of carbon nanotubes to be used, grams of adsorbed biophenols, grams of total adsorbate and the ratio, gbiophenol ads./ gtotal adsorbate, assuming a treatment of 1 liter of OVWs and operating with the conditions relating to systems 1, 2 and 3.”…“From data reported in Table 2 it is clear that using only 15 grams of carbon nanotubes and in 60 min, it is possible to recover 0.88 g of biophenols from 1 liter of OVWs. Data obtained are certainly interesting, considering also that the carbon nanotubes after a regeneration, for example with desorption in organic solvents, can be reused again. This makes the method simple and above all economically advantageous.

    1. For section 4: some problems present in this section, such as containing results, discussion; then I suggest the author rewrite the conclusion in one paragraph, which is different from the abstract.

R.14) The conclusions have been revised. Table 4 has been deleted and everything rewritten in a more concise way.

Reviewer 2 Report

The article is focused on using carbon nanotubes for treating olive vegetation waters.  The authors carried out experiments, which showed that the quantities of adsorbed biophenols depended on the quantities of carbon nanotubes used and the contact time.

However, I have a number of questions for the authors:

In Table 1, the authors refer to samples (systems) as 1, 2, and 3, but other names are used later in the text. Authors should give clear sample names and stick to them. The volume of OVWs was used for all tests - 20 ml. Column OVWs should be removed from Table 1.

Table 1 should be rewritten or removed as there is not any useful information. 

Figure 2 and 5 should be combined as figure 5 complement the information on figure 1. The data shown in these figures should be combined and presented in single figure

Can the authors confirm their hypothesis of agglomeration CNT formation by SEM or other methods?

Conclusions should be rewritten. What is the purpose of the authors presenting the data in Table 4?
What did authors want to demonstrate by proportionally increasing the quantity of nanotubes and solution?
Authors should move Table 4 to the Results part and provide a detailed explanation of the data presented or deleteTable 4.

What are the ways to increase the selective sorption of Biophenols?

Author Response

Dear colleague,

            first of all we thank you for the time you have committed to reviewing our work.

We have taken your suggestions into consideration and made adequate changes to the manuscript and answered and clarified our reasons.

Below are our answers following your suggestions (in the manuscript the changes are reported in yellow color).

Best regards

The authors

1) In Table 1, the authors refer to samples (systems) as 1, 2, and 3, but other names are used later in the text. Authors should give clear sample names and stick to them. The volume of OVWs was used for all tests - 20 ml. Column OVWs should be removed from Table 1. Table 1 should be rewritten or removed as there is not any useful information. 

R.1) Table 1 was removed from the manuscript. Also in paragraph 2.2 the following sentence has been added: “The adsorption tests were conducted in discontinuous mode and on different systems consisting of a constant volume of OVWs equal to 20mL and different quantities of carbon nanotubes equal to 0.3; 0.5 and 1.0 grams, respectively called system 1, 2 and 3. The adsorption times were 5, 10, 20, 45 ,60, 120 minutes.”

2) Figure 2 and 5 should be combined as figure 5 complement the information on figure 1. The data shown in these figures should be combined and presented in single figure

R.2) If this is not a problem for you, we will prefer to keep the two figures separate, as the first (Fig. 2) shows the total amount adsorbed while figure 5 shows the percentage of biophenols adsorbed. Furthermore, this would result in a substantial change in the organization of the manuscript.

3)Can the authors confirm their hypothesis of agglomeration CNT formation by SEM or other methods?

R.3) We did not consider it necessary to carry out further investigations because the agglomeration phenomenon is evident to the naked eye. For clarity the following sentence has been added in paragraph 3.2.: “An increase in the quantities of nanotubes, beyond a certain value, generates agglomeration phenomena of the nanotubes due to poor dispersion of the latter and visible to the naked eye after  few seconds from the start of the adsorption tests.”

4)Conclusions should be rewritten. What is the purpose of the authors presenting the data in Table 4? What did authors want to demonstrate by proportionally increasing the quantity of nanotubes and solution? Authors should move Table 4 to the Results part and provide a detailed explanation of the data presented or deleteTable 4.

R.4) The conclusions have been revised. Table 4 has been deleted and everything rewritten in a more concise way.

Table 4 (now Table 2) has been moved to paragraph 3.4. In addition, the following sentences have been added:

“From the data obtained it was possible to hypothesize a treatment of 1 L of OVWs in order to highlight more concretely which are, in the best experimental conditions found, the quantities of carbon nanotubes that must be used, and the quantities of biophenols that are recovered. The following Table 2 reports the grams of carbon nanotubes to be used, grams of adsorbed biophenols, grams of total adsorbate and the ratio, gbiophenol ads. / g total adsorbate, assuming a treatment of 1 liter of OVWs and operating with the conditions relating to systems 1, 2 and 3.”“From  data reported in Table 2 it is clear that using only 15 grams of carbon nanotubes and in 60 min, it is possible to recover 0.88 g of biophenols from 1 liter of OVWs. Data obtained are certainly interesting, considering also that the carbon nanotubes after a regeneration, for example with desorption in organic solvents, can be reused again.” This makes the method simple and above all economically advantageous.

5) What are the ways to increase the selective sorption of Biophenols?

R.5) The following sentence was added in the conclusions: “In summary, it is possible to state that a greater adsorption selectivity with respect to biophenols in OVWs is guaranteed by an adequate and weighted quantity of carbon nanotubes. An excess of the quantities of carbon nanotubes presupposes a lowering of selectivity.”

Round 2

Reviewer 1 Report

  1. Please add "respectively" after "120 minute" in Line 105;
  2. Please change the style in Line 284-289 and Line 303-308.

Author Response

Dear colleague,

again thank you for your time taking our review.

All the latter suggestions:

1)Please add "respectively" after "120 minute" in Line 105;

2) Please change the style in Line 284-289 and Line 303-308.

have been made and reported in the manuscript in yellow.

  Many greetings

The authors

Reviewer 2 Report

the authors have corrected some parts of the article in accordance with my comments so I recommend to publish the article in present form

Author Response

Dear colleague,

we thank you for your time that you have spent on our review.

Best regards

The authors